# Comparison of Cu-CHA-Zeolites in the Hybrid NSR-SCR Catalytic System for NO$_x$ Abatement in Mobile Sources

**Sergio Molina-Ramírez**, **Marina Cortés-Reyes**, **Concepción Herrera**, **María Ángeles Larrubia** and **Luis José Alemany** *

Departamento de Ingeniería Química, Facultad de Ciencias, Campus de Teatinos, Universidad de Málaga, E-29071 Málaga, Spain
* Correspondence: luijo@uma.es; Tel.: +34-952-131-919

**Abstract:** DeNO$_x$ activity in a NSR–SCR hybrid system of two copper-containing chabazite-type zeolitic catalysts was addressed. A Pt-Ba-K/Al$_2$O$_3$ model catalyst was used as the NSR (NO$_x$ storage and reduction) catalyst. For the SCR (selective catalytic reduction) system, two Cu-CHA zeolites were synthesized employing a single hydrothermal synthesis method assisted with ultrasound and incorporating Cu in a 2 wt.%, 2Cu-SAPO-34 and 2Cu-SSZ-13. The prepared catalysts were characterized, and the crystallinity, surface area, pore size, HR-TEM and EDX mapping, coordination of Cu ions and acidity were compared. The NH$_3$ storage capacity of the SCR catalysts was 1890 and 837 μmol NH$_3$·g$_{cat}^{-1}$ for 2Cu-SAPO-34 and 2Cu-SSZ-13, respectively. DeNO$_x$ activity was evaluated for the single NSR system and the double-bed NSR–SCR by employing alternating lean (3%O$_2$) and rich (1%H$_2$) cycles, maintaining a concentration of 600 ppm NO, 1.5% H$_2$O and 0.3% CO$_2$ between 200 and 350 °C. The addition of the SCR system downstream of the NSR catalyst significantly improved NO$_x$ conversion mainly at low temperature, maintaining the selectivity to N$_2$ above 80% and reaching values above 90% at 250 °C when the 2Cu-SSZ-13 catalyst was located. The total reduction in the production of NH$_3$ and ~2% of N$_2$O was observed when comparing the NSR–SCR configuration with the single NSR catalyst.

**Keywords:** DeNO$_x$ hybrid technology; NSR–SCR; Cu-CHA zeolites; SAPO-34; SSZ-13

## 1. Introduction

Air pollution remains a large environmental and health risk worldwide. Even though the industrial sector plays a role, road transport is still the major contributor. In fact, according to the European Environmental Agency (EEA), the road-transport sector is responsible for 43.31% of the harmful NO$_x$ emissions in Europe with passenger cars being the highest contributor [1]. Accordingly, the reduction of NO$_x$ emissions and tightening of policies that regulate NO$_x$ emissions from light-duty diesel vehicles have been carried out over the last few years. In fact, the latest proposal of the Euro 7 regulation, which will come into force in Europe in 2025, restricts NO$_x$ emission levels for light-duty internal combustion engine vehicles from 80 mg·km$^{-1}$ to 60 mg·km$^{-1}$ [2,3]. This context highlights the continuing importance of the development of DeNO$_x$ technologies and strategies and their direct application in the short to medium term. Indeed, the new Euro 7 regulation declares that technological progress should be reflected in lower emission limits that are based on state-of-the-art technologies, the combination of existing and current technologies and knowledge of pollution controls for all relevant pollutants [4].

For this purpose, NO$_x$ storage and reduction (NSR) and selective catalytic reduction (SCR) are two of the most effective and widely used technologies to efficiently remove nitrogen oxides from diesel exhaust gases. NSR systems consist of lean NO$_x$ trap (LNT) catalysts conformed by a high surface area supporting material ($\gamma$-Al$_2$O$_3$), an oxidation element (Pt, Pd or Rh) and NO$_x$-storage components, such as alkaline or alkaline-earth

metal oxides (Ba, K). This technology performs in two-cycled fed conditions, fuel–lean, which involves the oxidation of NO to $NO_2$ that is catalyzed by the noble metal and its storage as nitrate and nitrites on LNT components; then, during the subsequently short fuel-rich period, the stored nitrates and nitrites are reduced to the desired product $N_2$ and the undesired products $N_2O$ and $NH_3$-slip [5–8]. Meanwhile, SCR utilizes $NH_3$ (injected into the exhaust stream from an aqueous urea tank, *AdBlue*® Technology) as the $NO_x$ reductant, usually on a Cu- and/or Fe-exchanged zeolite-type catalyst, that leads to $N_2$ and $H_2O$ through the well-described SCR reactions [9–14].

In this sense, the coupling of NSR and SCR technologies is proposed as a plausible but still emerging solution in the combination of aftertreatment technologies in diesel vehicles [15–17]. The research is focused on combining the NSR systems with $NH_3$-SCR ones. The $NH_3$ formed on the NSR catalyst during the rich pulses can be used to increase the $NO_x$ removal efficiency of the NSR system if a $NH_3$-SCR catalyst is placed downstream. Thus, the ammonia is stored during the rich phase on the SCR catalyst and then used for the $NH_3$-SCR reactions in the next lean cycle. This hybrid technology would have the advantage of total or partial elimination of an external urea supply tank, as is currently the case in the single SCR system, favoring the $DeNO_x$ efficiency.

A NSR–SCR coupled system was studied and published in the literature that employed different and varied configurations of both catalytic materials. In addition, there are several patents on the coupling between LNT catalytic technology and a $NH_3$-SCR sample developed by companies such as Ford [18], BASF [19] and Johnson–Matthey [20]. In previous works published by our research group, the scaling from powder to the monolithic structure of the single-system conformation and the double-coupled after-treatment configuration was developed and deeply analyzed using a Pt-Ba-K/$Al_2O_3$ model catalyst as the NSR configuration followed by 2Cu-SAPO-34 as the $NH_3$-SCR system. An extensive parametric study under quasi-real conditions was presented that varied the operating conditions, space velocity and reducing agent during the rich period. In these works, we concluded that, at lab scale, the powder system was scalable to the monolithic configuration that presented a high $DeNO_x$ activity working in alternated cycles and was an optimal and improved conformation of the NSR–SCR hybrid system, keeping the volumetric ratio at 1:1 and a 2 wt.% in Cu content in the framework of the SAPO-34, synthesized with a one-pot hydrothermal method [16,21]. On the other hand, different NSR–SCR configuration studies are found in the literature. It is reported that copper-containing zeolites present higher $NO_x$ conversions than Fe catalysts for the NSR–SCR configuration, concluding that Cu catalysts result in improved performance for the combined double bed [22]. Furthermore, it is suggested that Cu-chabazite-type zeolites are the most promising SCR catalyst for this type of system due to their high thermal durability and selectivity to $N_2$ [23]. In fact, Cu-containing chabazite zeolites, such as silicoaluminophosphate SAPO-34 and high-silica SSZ-13, are currently considered and widely employed as highly efficient, excellent poison-resistant and robust SCR catalysts for $NO_x$ reduction in exhaust gases for internal combustion engines (ICE) [24].

Although there are a variety of studies, configurations and different materials employed for the analysis of this coupled-hybrid technology, the evaluation and knowledge of the development of SCR catalysts with Cu content are still important issues. This work is focused on the synthesis of two isostructural zeolites with copper integrated into their structure, SAPO-34 and SSZ-13 CHA, and the combination and analysis of the behavior in $DeNO_x$ing activity in the NSR–SCR hybrid system. Alternated pulses, switching between lean and rich conditions, were performed for understanding the $DeNO_x$ removal.

## 2. Materials and Methods

### 2.1. Catalyst Synthesis

#### 2.1.1. NSR Catalyst

For the NSR system, a deeply analyzed model catalyst developed in our research group for this process was employed [25–28]. The 0.4Pt-3.5Ba-1.5K/$\gamma$-$Al_2O_3$ catalyst was

prepared on alumina powder support with the wetness impregnation method of different precursors. The metal loading over the catalyst is expressed as the surface atomic density (at·nm$^{-2}$) in terms of surface coverage. The synthesis was carried out in two consecutive impregnations. First, the impregnation of $\gamma$-Al$_2$O$_3$ (Sasol Puralox TH, A$_{BET}$ = 144 m$^2$·g$^{-1}$ and V$_p$ = 1 cm$^3$·g$^{-1}$) with the platinum precursor of an aqueous solution of Pt(NH$_3$)$_2$(NO)$_2$ (3.4 wt.%, Aldrich Chemical, Schnelldorf, Germany) was carried out. After drying in a static air oven overnight at 80 °C and calcined at 350 °C (5 °C·min$^{-1}$) for 3 h, the pure alkaline precursors Ba(CH$_3$COO)$_2$ (Merck, Darmstadt, Germany) and KC$_2$O$_2$H$_3$ (Fluka, Morriston, NJ, USA) were incorporated with the same procedure as a mixed adequate aqueous solution and left drying overnight at 80 °C with static air. The final calcination was performed at 500 °C with a ramp of 5 °C·min$^{-1}$ for 5 h in air.

### 2.1.2. SCR Catalysts

Isostructural chabazite-type zeolitic materials were synthesized with a one-pot method developed in our research group with Cu incorporation. SAPO-34-type materials were prepared with a hydrothermal method assisted with ultrasound and improved by our research group in previous work for CHA-type zeolites [29]. The molar gel composition was 2 DEA: 0.6 SiO$_2$: 1 Al$_2$O$_3$: 0.8 P$_2$O$_5$: 50 H$_2$O. The preparation was carried out as follows. First, an aluminum suspension was prepared by adding aluminum isopropoxide (Alfa Aesar, Kandel, Germany) to water (75 wt.%); then, distilled water and phosphoric acid (Panreac) were incorporated and vigorously stirred for 2 h. After that, a silica source as collodial silica (LUDOX$^{®}$ HS-40, Sigma Aldrich, Burlington, MA, USA) was added to the solution for 4 h. Finally, the structure-directing agent (diethylamine, DEA, supplied by Sigma Aldrich) and 5 wt.% of commercial SAPO-34 seeds supplied by Albemarle were included. Before the transferring of the precursor solution, it was subjected to sonication for 5 min with pulses of 30 s at room temperature (Ultrasonic Processor UIP1000hd Hielscher with an unchangeable frequency of 20 kHz using a titanium sonotrode and an ultrasonic power intensity of 300 W·cm$^{-2}$). The precursor gel was autoclaved at 200 °C for 4 h. The solid product was washed several times with distilled water, filtered and dried overnight at 110 °C. Finally, it was calcined at 550 °C (2 °C·min$^{-1}$) in air for 5 h to remove the organic template molecules.

SSZ-13 materials were also synthesized with a hydrothermal method following the conventional procedure described in the literature [30] and improved with an ultrasound-assisted method, similar to that for SAPO-34 as described above. The molar gel composition was 10 Na$_2$O: 2.5 Al$_2$O$_3$: 100 SiO$_2$: 4400 H$_2$O: 20 RN-OH. First, a mixture of the appropriate amount of precursors was prepared: a 1 M solution of NaOH (PanReac AppliChem, Barcelona, Spain), the structural directing agent *N,N,N*-trimethyl-1-adamantylammonium (RN-OH) supplied by TCI Chemicals, and a small amount of distilled water. The solution was allowed to stabilize under stirring for 4 h. Subsequently, the required molar amount of Al(OH)$_3$ (Sigma Aldrich) was gradually added to the solution and stirred until the solution was clear. Then, the SiO$_2$ source as LUDOX$^{®}$ was incorporated dropwise under ambient conditions. A total of 1 wt.% of SAPO-34 seeds was incorporated to the mixture due to the CHA-structural improvements provided. Finally, the precursor solution was kept under vigorous stirring for 4 h and subjected to sonication for 5 min with pulses of 30 s at room temperature. Later, the solution was transferred to a Teflon-lined autoclave and kept isolated at 160 °C in an oven for 5 days. The resulting solid was washed with distilled water several times, recovered via centrifugation, dried at 80 °C for 24 h, and finally calcined at 600 °C (2 °C·min$^{-1}$) in air for 5 h to remove the organic template molecules.

Copper was introduced into both the SAPO-34 and SSZ-13 zeolites during the one-pot synthesis process with a metal loading of 2 wt.% (expressed as copper metal). For this purpose, a copper complex was prepared by mixing a 20 wt.% solution of copper (II) sulfate with tetraethylenepentamine (TEPA, Sigma Aldrich) and kept under stirring and stabilization for 4 h until the solution was complete. The solution was added to the

precursors' solution and kept in stirring for 5 h. The copper–CHA catalysts were denoted as 2Cu-SAPO-34 and 2Cu-SSZ-13.

All the catalyst samples were milled and sieved (100-80 MESH) to avoid mass transfer limitations during the characterization techniques and reactivity analysis.

## 2.2. Catalyst Characterization

X-ray diffraction (XRD) patterns were collected using an X'Pert MPD Pro diffractometer (PANanalytical) equipped with Cu Kα radiation (λ = 1.5418 Å).

The specific surface areas ($m^2 \cdot g^{-1}$) for the zeolite samples were determined using the Brunauer–Emmet–Teller (BET) model on a Micromeritics ASAP 2020 Analyzer. Before the analysis, the samples were outgassed in a vacuum ($1 \times 10^{-3}$ Pa) at 80 °C, and then nitrogen adsorption and desorption isotherms were obtained at liquid nitrogen temperature.

High-resolution transmission electron microscopy (HR-TEM) images were obtained in a TalosTM F200X instrument equipped with an energy dispersive X-ray (EDX) analyzer to obtain an elemental analysis mapping.

Diffuse reflectance (DR) UV-Vis analysis was carried out in air in wet conditions using a TP92 + UV spectrophotometer with a conventional integrating sphere.

Temperature-programmed desorption (TPD) of the preadsorbed $NH_3$ was performed in an AutoChem 2920 (Micromeritics) instrument with a TCD detector to study surface acidic sites in the zeolite samples. Before the $NH_3$–TPD measurements, the samples were conditioned at 500 °C in a helium flow for 1 h and subsequently cooled to 100 °C. Then, the catalysts were exposed to $NH_3$ at 100 °C for 30 min followed by a temperature ramp from 60 °C at a temperature rate of 30 °C·$min^{-1}$ up to 800 °C.

## 2.3. Catalytic Activity
### NSR Alternating Cycles (DeNO$_x$ Activity)

The catalytic activity of the single NSR and hybrid NSR–SCR was evaluated using the transient response method (TRM). For this study, catalytic powder samples (60 mg) were placed into a U-type quartz reactor ($d_i$ = 6 mm). For the NSR–SCR configuration, the same amount of each catalyst was placed consecutively in the reactor: two catalytic beds of 60 mg each (*w/w*: 1/1) separated by a quartz wool layer. For the single NSR tests, the total weight of the double bed and same configuration were maintained, replacing the SCR bed with an inert quartz. All samples were pretreated in helium carrier flow maintaining flow-reaction conditions up to 500 °C for 1 h. After that, lean ($t_L$ = 150 s) and rich ($t_R$ = 20 s) cyclings were alternated in the temperature range between 200 and 350 °C under constant flow, consisting of 1.5% of $H_2O$ and a 0.3% $CO_2$ atmosphere, using He as a carrier and maintaining the GSHV = 30,000 $h^{-1}$ (100 mL·$min^{-1}$) calculated with respect to the NSR catalytic bed volume. NO was introduced during both cycles, and the concentration was maintained at 600 ppm. During the oxidizing period (lean cycle), 3% of oxygen was fed, whereas 1% of hydrogen was introduced into the feeding stream during the reductant period (rich cycle). The exhaust gas and the product distribution were monitored online utilizing a MultiGas 2030 FTIR gas analyzer, and a Pfeiffer Prisma$^{TM}$ mass spectrometer QMS 200 was employed in order to follow the $N_2$, $H_2$ and $O_2$ signals. From the mathematical treatment and the quantitative analysis of the product distribution profiles of an average of three stable cycles, $NO_x$-conversion (X) and selectivity (S) values were obtained through the following equations:

$$X_{NO_x}(\%) = \frac{A_{NO_x}^{in} - A_{NO_x}^{out}}{A_{NO_x}^{in}} \cdot 100 \tag{1}$$

$$S_{N_2}(\%) = \frac{2 \cdot A_{N_2}^{out}}{A_{NO_x}^{in} - A_{NO_x}^{out}} \cdot 100 \tag{2}$$

$$S_{NH_3}(\%) = \frac{A_{NH_3}^{out}}{A_{NO_x}^{in} - A_{NO_x}^{out}} \cdot 100 \tag{3}$$

$$S_{N_2O}(\%) = \frac{2 \cdot A_{N_2O}^{out}}{A_{NO_x}^{in} - A_{NO_x}^{out}} \cdot 100 \tag{4}$$

where A represents the inlet and outlet areas.

## 3. Results

### 3.1. Characterization

The lean $NO_x$ trap (LNT) model catalyst 0.4Pt-3.5Ba-1.5K/$Al_2O_3$ is presented in the literature as a high $DeNO_x$ and PM-removal performance material [26,28,31]. In Figure 1, the XRD pattern of Pt-Ba-K is shown. The signals associated with the characteristic structure of $\gamma$-alumina (JCPDS 75-0921) were observed, and platinum in the metallic state (JCPDS 4-0802) was identified. Witherite, as the orthorhombic carbonate species, was the Ba-dominant phase (JCPDS 5-0378). The peaks assigned to potassium oxide ($K_2O$) were also observed (JCPDS 43-1020) with no detection of the K-hydrated carbonated species. More detailed properties and characterization techniques, including X-ray photoelectron spectroscopy (XPS) and HR-TEM, are reported in previous works [26–28,32].

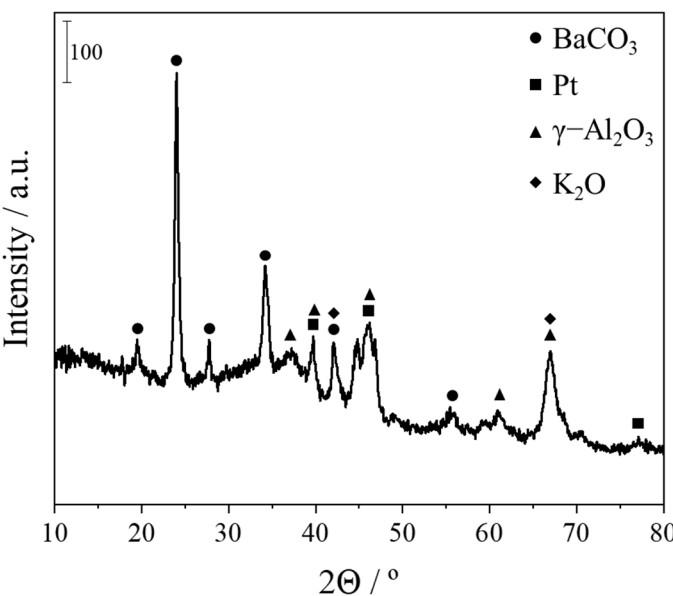

**Figure 1.** X-ray diffraction pattern of the NSR catalyst 0.4Pt-3.5Ba-1.5K/$Al_2O_3$.

Focusing henceforth on the results of the characterization of the zeolitic materials, the XRD diffractograms of both Cu zeolites after the calcination process are shown in Figure 2. All the observed peaks (2θ = 9.4, 12.9, 16.15, 20.5, 26.1 and 30.7°) and the complete diffraction patterns belonged to the chabazite topological structure (JCPDS 01-087-1527) for both samples. The unit cell parameters a, b and c, and the volume calculated with Rietveld refinement of both samples (Table 1) were similar and slightly higher than, due to the incorporation of copper [33], those reported in the literature for the simple chabazite phase [34]. Otherwise, a higher crystallinity was observed for 2Cu-SSZ-13 compared to that of the Cu-SAPO sample (an increase of 11.4% in crystallite size). In the literature, it is reported that the Cu–TEPA complex is an effective and special organic structure-directing agent (OSDA) for synthesizing CHA-type zeolites, specifically for the SSZ-13 one-pot synthesis method, and improves the crystallinity and catalytic performance in the $NH_3$-SCR reaction [35,36]. The X-ray diffractograms associated with 2Cu-SAPO-34 (Figure 2a) and 2Cu-SSZ-13 (Figure 2b) did not show peaks at 35.4 and 38.2° attributable to CuO or

any other feature assignable to the $CuO_x$ species, which suggests that copper ions can be uniformly distributed and integrated into the lattice of the CHA framework or highly dispersed.

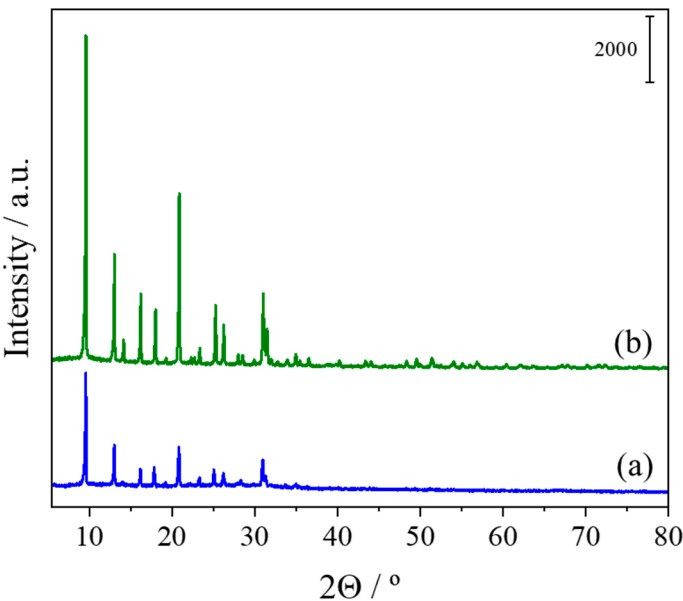

**Figure 2.** X-ray diffraction patterns of the SCR Cu samples (**a**) 2Cu-SAPO-34 and (**b**) 2Cu-SSZ-13.

**Table 1.** Surface area, pore volume and cell unit parameters of the Cu zeolites.

| Catalyst | Surface Area ($m^2 \cdot g^{-1}$) | | | Pore Volume ($cm^3 \cdot g^{-1}$) | | Cell Unit Parameters * | | | |
|---|---|---|---|---|---|---|---|---|---|
| | $S_\mu$ | $S_{ext}$ | $S_{BET}$ | $V_\mu$ | $V_{total}$ | V (Å³) | a = b (Å) | c (Å) | Cryst. Size (Å) |
| 2Cu-SAPO-34 | 435 | 85 | 520 | 0.142 | 0.369 | 2381.51 | 13.589 | 14.892 | 1144.7 |
| 2Cu-SSZ-13 | 793 | 13 | 806 | 0.279 | 0.294 | 2352.15 | 13.564 | 14.761 | 1292.4 |

\* calculated with Rietveld refinement.

The calculated results of the surface area and the pore volume from the $N_2$ adsorption–desorption isotherms are shown in Table 1. Both the 2Cu-SAPO-34 and 2Cu-SSZ-13 samples presented isotherms that closely resembled type I according to the Brunauer–Deming–Deming–Teller (BDDT) classifications that correspond to microporous materials. The surface area and total pore volume of 2Cu-SAPO-34 and 2Cu-SSZ-13 showed close values that were previously reported [37–44]. It can be observed that the total surface area of the 2Cu-SSZ-13 material was considerably higher than that of the 2Cu-SAPO-34 material. Meanwhile, the ratio between the microporous surface ($S_\mu$) and the surface area ($S_{BET}$) showed high values for both materials: $S_\mu/S_{BET} = 0.98$ and 0.84 for 2Cu-SSZ-13 and 2Cu-SAPO-34, respectively. This suggests that both materials maintain a mainly microporous structure. On the other hand, attending to the $V_\mu/V_{total}$ ratio, a value of 0.38 was observed for the Cu-modified SAPO-34 and 0.95 for 2Cu-SSZ-13, which indicates a higher mesopore volume size for the 2Cu-SAPO-34 zeolite, while the 2Cu-SSZ-13 microporous structure and mainly micropore volume were confirmed.

In order to study and better understand the dispersion and distribution of copper in the sample, HR-TEM images and EDX mapping were recorded and are displayed in Figure 3. In all samples, the typical shape of chabazite-type zeolites was observed with a slight difference between 2Cu-SAPO-34 (Figure 3a) and 2Cu-SSZ-13 (Figure 3b) [21,41,45,46]. In addition, the particle size presented a uniform value between 3 and 4 nm in the two investigated samples. The homogeneous distribution of all elements present in the zeolitic

samples, such as silica and aluminum and, in addition, phosphorus in the case of 2Cu-SAPO-34, was confirmed from the images extracted with EDX mapping. Moreover, focusing on the detected copper, it was possible to confirm the evident high dispersion of the element in all particles of both Cu zeolites without the presence of localized clusters. The results obtained from the average composition with the EDX analysis for both zeolites, presented in Table 2, coincided with their corresponding theoretical synthesis gel composition. The materials presented a silicon–aluminum ratio (Si:Al, SAR) equivalent to 0.23 and 18.67 for 2Cu-SAPO-34 and 2Cu-SSZ-13, respectively, which was within the margins of error with respect to the theoretical starting gel ratio composition of 0.3 and 20, respectively. It should be considered that, although both zeolites have the same chabazite structure, SSZ-13 is considered a total silica zeolite with a very low Al content. In addition, the phosphorus content for 2Cu-SAPO-34 was 19.67 wt.% and zero in the case of Cu-SSZ-13, as expected, a value practically similar to that previously declared with X-ray fluorescence (XRF) in previous published works [37]. Concerning the copper content in the two materials, the average value obtained was approximately 2 wt.%, indicating the correct incorporation of the element in mass percentage in relation to the starting value of the precursor gel previously calculated.

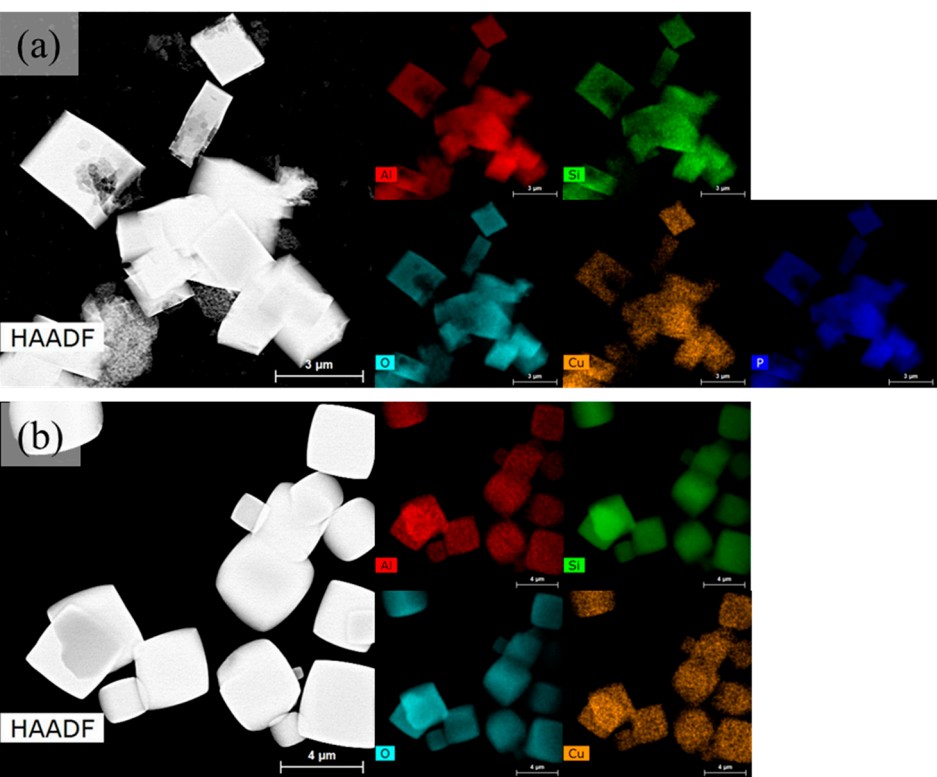

**Figure 3.** HR-TEM images and EDX mapping of the Cu-containing zeolites (**a**) 2Cu-SAPO-34 and (**b**) 2Cu-SSZ-13.

**Table 2.** Composition obtained with EDX mapping of HR-TEM.

|  | Al (wt.%) | Si (wt.%) | O (wt.%) | Cu (wt.%) | P (wt.%) |
|---|---|---|---|---|---|
| 2Cu-SAPO-34 | 25.75 | 6.17 | 46.45 | 1.95 | 19.67 |
| 2Cu-SSZ-13 | 2.88 | 55.95 | 39.10 | 2.07 | - |

The UV–Vis for the Cu-free SAPO-34 and SSZ-13 materials did not show absorption, and the spectra for the Cu-CHA zeolites are displayed in Figure 4. A main intense signal was recorded at 205 and 209 nm for the Cu-SSZ-13 and Cu-SAPO-34 modified zeolites, respectively. Cu-SSZ-13 also showed a shoulder at 280 nm that probably is due to the CuO

nano-differentiated species, while Cu-SAPO presented a main absorbance signal that was less symmetric and broader due to $Cu^{2+}$ speciation in the framework of the zeolite. A wide absorption band in the 750–850 nm range with a maximum at 825 nm (maximized in the upper-right part of Figure 5), also related to the presence of $Cu^{2+}$ species that are involved in $NH_3$-SCR activity [39,47,48], was observed. The most energetic signal at approximately 205–210 nm is attributed to the $O^{2-} \rightarrow Cu^{2+}$ ligand-to-metal charge transfer transition of isolated $Cu^{2+}$ species as reported in the literature for SAPO-34- and SSZ-13-type zeolites with Cu content [37,49,50]. The localized signal at approximately 750–850 nm is attributed to the d–d transition $2E_g \rightarrow 2T_{2g}$ signal originating from isolated $Cu^{2+}$ ions in an octahedrally coordinated $CuO_x$ environment [46,50,51]. By means of the analysis with XPS (Table 3), it was observed that the surface copper in the framework of the zeolitic materials presents differences in its coordination state depending on the CHA sample, maintaining a $Cu^+/Cu^{2+}$ ratio of 1.69 and 3.11 for 2Cu-SAPO-34 and 2Cu-SSZ-13, respectively, so that Cu would be mainly in the $Cu^+$ form, especially in SSZ-13 containing 2 wt.% of copper.

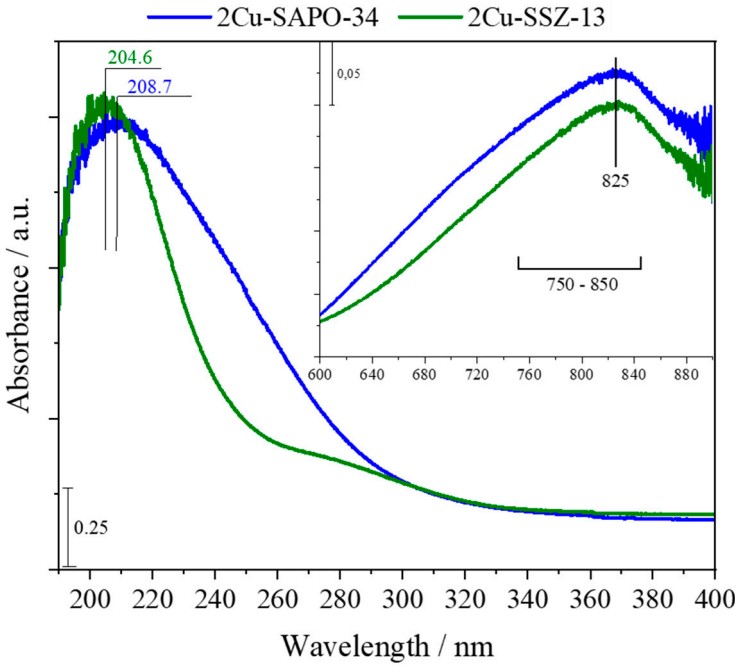

**Figure 4.** Diffuse reflectance UV–Vis spectra of the 2Cu-SAPO-34 and 2Cu-SSZ-13 samples.

**Table 3.** XPS results for the Cu-CHA zeolite samples.

| Sample | | Binding Energy (eV) | $Cu^+/Cu^{2+}$ Ratio |
|---|---|---|---|
| 2Cu-SAPO-34 | $Cu^+$ | 955.98 | 1.69 |
| | $Cu^{2+}$ | 954.96 | |
| 2Cu-SSZ-13 | $Cu^+$ | 953.05 | 3.11 |
| | $Cu^{2+}$ | 954.88 | |

Since the acidity in zeolitic catalysts is a significant influence on catalytic activity, in order to examine the variation of acidity and to better understand the active sites involved in SCR-catalysts during cycling lean–rich periods, $NH_3$-TPD tests were performed when the chemisorbed $NH_3$ was eventually released in both Cu-CHA samples. Figure 5 shows the ammonia-TPD profiles for 2Cu-SAPO-34 and 2Cu-SSZ-13. The TPD curves exhibited the typical "dual-acid sites" shape of $NH_3$ desorption for both Cu-chabazite SCR catalysts with a maximum located at 185 °C, which was associated with the weak acid sites, and

a second signal extended to 500 °C, which was associated with two contributions of the strong acid sites.

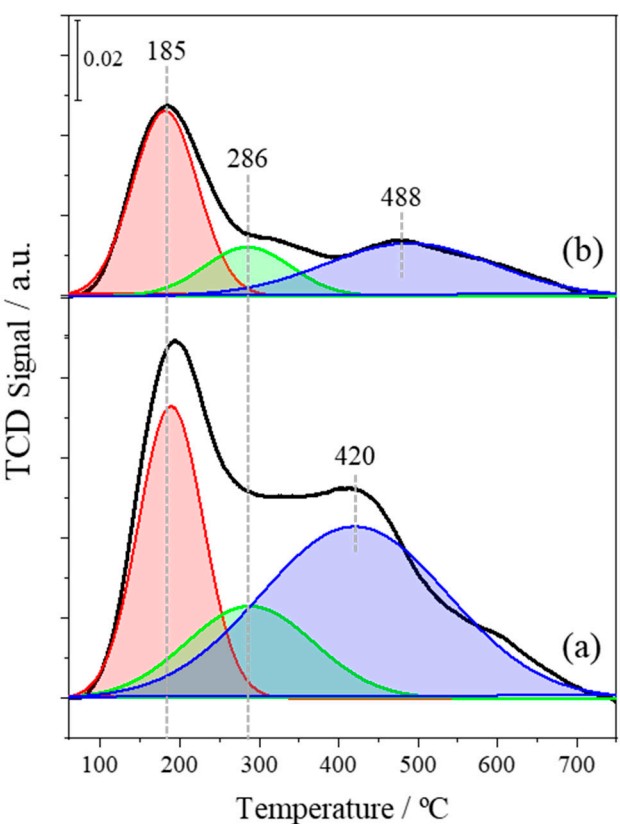

**Figure 5.** NH$_3$-TPD profiles for the Cu-CHA zeolites (**a**) 2Cu-SAPO-34 and (**b**) 2Cu-SSZ-13.

Three NH$_3$ desorption peaks at low temperature, 185 °C, attributed to weakly adsorbed NH$_3$ on terminal Si(Al)-OH as Brønsted acid sites (BAS). The intensity of the signal at 185 °C of Cu-SAPO-34 is stronger than that of Cu-SSZ-13, revealing that this catalyst presents more Si(Al)-OH groups. Thereafter, the second broad NH$_3$ desorption signal was identified at a temperature range between 250 to 500 °C, which was associated with two contributions of stronger acid sites in agreement to that reported by R. Villamaina et al. [52]. A medium-temperature signal due to NH$_3$ adsorbed on strong LAS, associated with isolated Cu$^{2+}$ sites, was identified, which increases the Lewis acidity localized at 286 °C for both Cu zeolites but was more evident for the Cu-SAPO-34 zeolite. High T-range desorption signals at 420 °C for 2Cu-SAPO-34 and 490 °C for 2Cu-SSZ-13 were observed; these last ammonia desorption signals must be related to the zeolite-framework Brønsted sites. The existence of different Cu species may cause the shift to the T-max desorption signal of the Brønsted sites.

Focusing on the two NH$_3$-TPD profiles obtained for both materials, the significant increase in acidity for 2Cu-SAPO-34 (Figure 5a) with respect to 2Cu-SSZ-13 (Figure 5b) was remarkable, being that the total amount of desorbed ammonia was 1890 and 837 µmol NH$_3 \cdot g_{cat}^{-1}$, respectively, which was, in part, related to the presence of phosphorous and a higher amount of Al in SAPO-34 compared to the "Si-enriched zeolite" SSZ-13 with a higher ammonia retention capacity at high temperature related to strong acid sites associated with the zeolitic framework. Furthermore, the overall acidity and the ratio between the Brønsted and Lewis acid sites in the Cu-chabazite zeolites depend slightly on the presence of Cu ions in agreement to that reported by C. Paolucci et al. [53]. So, the fist signal in the strong acid sites range was appreciable for both zeolites at approximately 286 °C, showing a higher relative contribution for 2Cu-SAPO-34 associated with the existence of different

populations of Cu cations in the zeolite structure that affect the overall $NH_3$ adsorption capacity of the Cu-modified zeolites.

### 3.2. DeNO$_x$ Activity

The removal performance was studied by alternating pulses that switched between lean and rich conditions for two Cu-chabazite-type zeolitic catalysts, SAPO-34 and SSZ-13, as the SCR catalyst forming part of the hybrid NSR–SCR catalytic system.

$NO_x$ conversion and the selectivity towards $N_2$, $NH_3$ and $N_2O$ of the single 0.4Pt-3.5Ba-1.5K/Al$_2$O$_3$ system (NSR) and the NSR–SCR double-bed configuration, Pt-Ba-K/Al$_2$O$_3$+ 2Cu-SAPO-34 or Pt-Ba-K/Al$_2$O$_3$+2Cu-SSZ-13, were studied as a function of the operating temperature (Figure 6). For the single NSR catalytic bed, an increase in the conversion of nitrogen oxides with the temperature was observed from 48% to 59% at 200 and 350 °C, respectively. This trend coincided with the increase in $N_2$ selectivity achieved from 48% at the lower temperature to approximately 97% at the highest temperature range analyzed. On the contrary, a progressive reduction in $NH_3$-selectiviy values was observed, decreasing with increasing temperature and reaching values of 1.5% at 350 °C, while $N_2O$ selectivity decreased from 16.7 to 1.9%. These results were in consonance with those previously published for the Pt-Ba-K system in monolithic form maintaining the NO inlet during the rich and lean periods under $H_2O$+$CO_2$ atmosphere conditions [16].

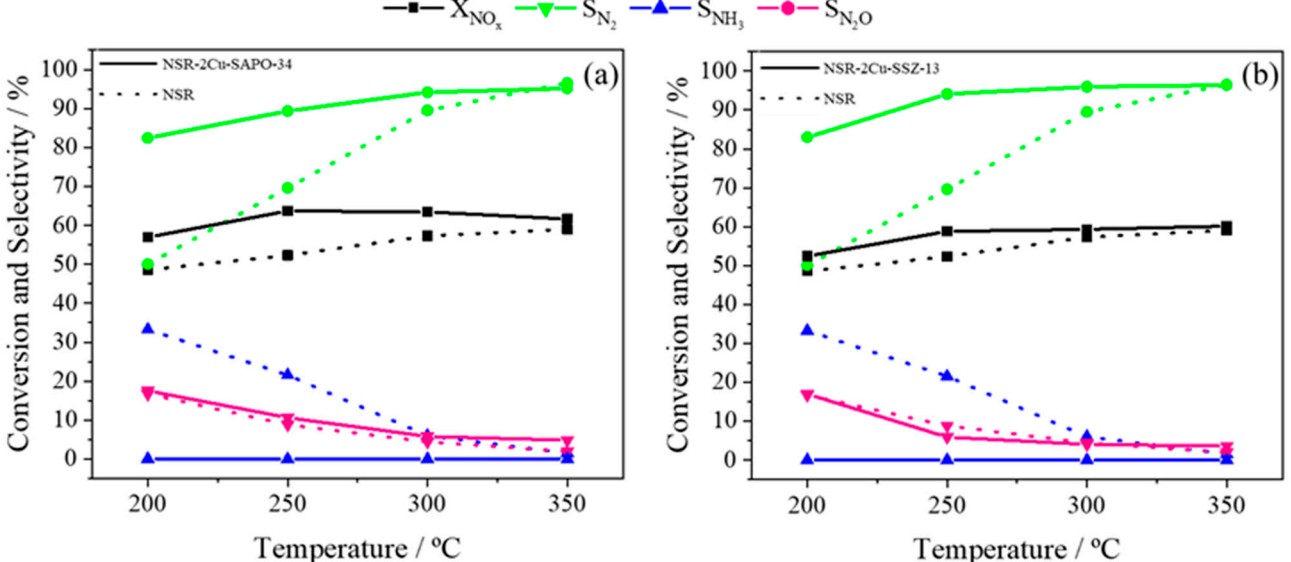

**Figure 6.** DeNO$_x$ activity performance for 0.4Pt-3.5Ba-1.5K/Al2O3 as the NSR system (dotted line) and hybrid NSR–SCR coupled catalytic configuration (solid line): (**a**) NSR-2Cu-SAPO-34 and (**b**) NSR-2Cu-SSZ-13.

Concerning the catalytic activity results for the hybrid NSR–SCR configurations, it was noted that the $NO_x$-conversion values were higher than those observed for the single NSR system throughout the temperature range; however, this increase was notably greater at the low temperature range since at 350 °C the calculated $X_{NO_x}$ values were almost similar to those registered for the single NSR catalyst due to the low amount of ammonia formed in the Pt-Ba-K catalyst at high temperatures. The conversion values increased by approximately 10% at 300 °C for both coupled systems with respect to the single NSR system, which led to a significant increase in $N_2$ selectivity. For the NSR+2Cu-SAPO-34 hybrid catalytic system (Figure 6a), the $NO_x$ conversion values were higher than those calculated for the NSR+2Cu-SSZ-13 hybrid catalytic system (Figure 6b), which was associated with the higher overall $NH_3$ adsorption capacity and $Cu^{2+}$ sites population.

The NSR–SCR hybrid technology maintained a higher $S_{N_2}$ than that calculated for the single NSR configuration in the whole range tested. It was remarkable that the increase in $N_2$ selectivity values between 250 and 300 °C, which are maintained at higher temperatures, reached a maximum close to 96% at 350 °C for both SCR catalysts. For the NSR+2Cu-SSZ-13 configuration, higher $N_2$ selectivity values of 94% were reached at 250 °C and above, while the NSR+2Cu-SAPO-34 system showed values close to 90% at this temperature. It is assumed that the improvement in nitrogen selectivity and $NO_x$ abatement for the hybrid NSR–SCR system compared to the single NSR bed is due to the fact that the ammonia generated at the reducing NSR-cycling period by the Pt-Ba-K catalyst is constantly retained in the consecutive zeolitic material and acts as an efficient reductant, favoring the reduction of $NO_x$ to $N_2$ by the SCR reactions and keeping the $NH_3$ concentration values close to zero at the reactor outlet. Nevertheless, as could be observed in the selectivity values of the single NSR configuration, the low ammonia production at temperatures above 300 °C (values close to 3%) resulted in a lower or not appreciable increase in $NO_x$ conversion efficiency and $N_2$ selectivity for the two coupled NSR–SCR configurations.

Regarding the $N_2O$ selectivity, the values calculated for the NSR and NSR–SCR systems were similar. In fact, a progressive decrease with the temperature was observed from approximately 16% to values lower than 2% between 200 and 350 °C, respectively. $N_2O$ is generated during the rich period and at the rich/lean transition due to the combination between the incomplete reduction of $NO_x$ adsorbed on the NSR catalyst at low temperature and the exothermic reaction between NO and $H_2$ at the rich period at high temperature [16,54]. On the other hand, lower $N_2O$-selectivity values were noted at temperatures below 250 °C for the NSR-2Cu-SSZ-13 configuration compared to the single NSR system. This phenomenon was not observed in the case of the NSR-2Cu-SAPO-34 coupled system and is attributed to the ability to reduce $N_2O$ production due to its decomposition by the copper ionic pairs or clusters present in the zeolitic structure [12]. Therefore, although both SCR catalysts improve $NO_x$ conversion values, which increases nitrogen selectivity, the influence of Cu-modified SSZ-13 in the occurrence of parallel reactions is more suitable for its use in the hybrid catalytic system.

## 4. Conclusions

In the present work, two zeolite-type catalysts with chabazite structure and 2 wt.% copper content were developed with the one-step hydrothermal method to evaluate their properties, and the influence of adding a CHA zeolite–SCR catalyst (two catalytic beds upstream Pt–Ba-K/$Al_2O_3$ and downstream 2Cu-SAPO-34 or 2Cu-SSZ-13) on the $NO_x$ removal activity of the NSR+SCR hybrid catalytic system was analyzed.

The $NO_x$ removal activity with the single NSR technology under an alternating lean–rich atmosphere had different impacts in terms of selectivity by the incorporation of the Cu-SAPO-34 and Cu-SSZ-13 zeolites. There is an enhanced activity in the $NO_x$ removal for both the Pt–Ba-K/$Al_2O_3$+Cu-SAPO-34 catalyst and Pt–Ba-K/$Al_2O_3$+Cu-SSZ-13 catalyst hybrid technologies using NSR conditions in the presence of water and $CO_2$ and $H_2$ as the reductant compared to the single NSR catalyst.

Both materials presented the same chabazite structure and copper content (although different SAR). The environment and coordination state of the copper species, in spite of the overall acidity of Cu-SAPO-34, are higher, and they influenced their catalytic activity. The Cu-CHA catalysts showed a total reduction in the ammonia produced by the NSR catalyst under the conditions studied, storing the ammonia and acting as a $NO_x$ reductant to $N_2$ through the SCR reactions. The 2Cu-SSZ-13 catalyst presented a lower acidity and a lower $Cu^{2+}$ ion population in correlation with the increase in $Cu^+$ ions. The obtained values of activity in the NSR–SCR double bed were slightly improved in terms of selectivity to $N_2$ (~5%) in the lower temperature range. Above 300 °C, ammonia production was negligible for the NSR catalyst, and the influence of placing a downstream SCR zeolite was not significant; therefore, the behavior of both zeolites was similar.

**Author Contributions:** S.M.-R.: Methodology, investigation, and writing—review. M.C.-R.: Investigation and writing—review. C.H.: Investigation, conceptualization, and writing—review. M.Á.L.: Investigation and writing—review. L.J.A.: Conceptualization, resources, and writing—review and editing, writing-review, and supervision. All authors have read and agreed to the published version of the manuscript.

**Funding:** Economy and Competitiveness Spanish Ministry CTQ 2017-87909-R project. S.M.R.'s Ph.D. Grant (PRE-2018-086107) funded by Economy and Competitiveness Spanish Ministry.

**Institutional Review Board Statement:** Not applicable.

**Informed Consent Statement:** Not applicable.

**Data Availability Statement:** Data will be made available on request.

**Acknowledgments:** SMR acknowledges the Economy and Competitiveness Spanish Ministry for the Ph.D. grant (PRE-2018-086107). Authors want to thank the financial support of CTQ 2017-87909-R Project from the Economy and Competitiveness Spanish Ministry. PROCAT research group (Chemical Engineering Department at the University of Málaga) wishes to thank the invitation to participate in this Special Issue dedicated to the scientific career of Corma and also wants to express our deepest appreciation to Avelino Corma.

**Conflicts of Interest:** The authors declare no conflict of interest.

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
