# Peer review of "Comparison of Cu-CHA-Zeolites in the Hybrid NSR-SCR Catalytic System for NOx Abatement in Mobile Sources"

_chemistry, doi:10.3390/chemistry5010043_

Round 1
Reviewer 1 Report
The article covers very popular topic - NOx abatement by Cu-containing zeolites. The main idea of the article is to combine SCR and NSR strategies for deNOx, which is very interesting. A big part of the article covers the characterization of SSZ-13 and SAPO-34 zeolites with 2wt% of Cu. They are characterized with XRD, TEM, EDX, UV-VIS, TPD. But then, probably the most important results, are presented in Figure 6, where the catalytic activity of the industrial catalyst for NSR and the mixture of it with Cu-zeolites are compared.
So, my main question is why authors did not characterize the mixture of NSR and Cu-zeolites the same way as zeolitic samples (physicochemical charaterization: XRD, TEM, etc.)? Or why a pure Cu zeolitic phase was not examined in catalysis. With these parts, the article would have a more reasonable structure.
Where is the legend to Figure 6 (colors)?
In my opinion, abbreviations should be exaplained in the abstract (like NSR and SCR), there are some spelling mistakes in the article. It should be noted in the Introduction what is the difference between SAPO-34 and SSZ-13.
Author Response
Reviewer 1
The article covers very popular topic - NOx abatement by Cu-containing zeolites. The main idea of the article is to combine SCR and NSR strategies for deNOx, which is very interesting. A big part of the article covers the characterization of SSZ-13 and SAPO-34 zeolites with 2wt% of Cu. They are characterized with XRD, TEM, EDX, UV-VIS, TPD. But then, probably the most important results, are presented in Figure 6, where the catalytic activity of the industrial catalyst for NSR and the mixture of it with Cu-zeolites are compared.
- So, my main question is why authors did not characterize the mixture of NSR and Cu-zeolites the same way as zeolitic samples (physicochemical charaterization: XRD, TEM, etc.)? Or why a pure Cu zeolitic phase was not examined in catalysis. With these parts, the article would have a more reasonable structure.
The NSR-SCR hybrid technology consists of the coupling of two catalytic systems widely used for NOx abatement in diesel engines, the NOx Storage and Reduction (NSR) system and the Selective Catalytic Reduction (SCR) system. The coupling between a NOx-Trap followed by a NH3-SCR catalyst leads to an improved N2-selectivity, by the storage and reaction of the undesired NH3 produced in the NSR catalyst in the downstream SCR system. The NSR catalyst 0.4Pt-3.5Ba-1.5K/Al2O3 has been extensively characterized in previous publications of our research group (see references 25-28) for NOx Storage and Reduction process. In turn, the 2Cu-SAPO-34 catalyst for the NH3-SCR system has been optimized in previous studies (see references 12 and 39). As both are consecutively coupled independent systems, a characterization study of the mixture of both is not considered necessary.
- Where is the legend to Figure 6 (colors)?
This was corrected in the Manuscript.
(line 357)
- In my opinion, abbreviations should be exaplained in the abstract (like NSR and SCR), there are some spelling mistakes in the article. It should be noted in the Introduction what is the difference between SAPO-34 and SSZ-13.
The meaning of the main abbreviations of the NOx Storage and Reduction (NSR) and Selective Catalytic Reduction (SCR) system is given in the abstract. Subsequently, in the Introduction section, we consider that both technologies processes were deeply described and explained.
SAPO-34 and SSZ-13 materials belongs to the chabazite zeolites group. The CHA framework consists of double six-membered rings (D6R) stacked in an ABC-sequence and interconnected with four membered rings (4MR) which are interconnected with eight-membered ring (8MR) windows. SAPO-34 contains phosphorous (silicoaluminophosphate) and SSZ-13 is composed of a high-silica content CHA-zeolite. It was included at line 82.

Reviewer 2 Report
1. Full names need to be given to abbrevations such as BDDT.
2. What is the reason for Ba in BaCO3 but K in K2O from Fig1, when K(I) is more basic than Ba(II) and easier to form carbonate with CO2.
3. No XPS results was shown as claimed. It would be good if there was Cu XPS information.
4. The legend in Fig 6 is not clear.
5. What is the main reason for the peak at 488 degC for Cu-SSZ-13 and at 420 for Cu-SAPO-34? What is the effect of stronger acid for Cu-SSZ-13 in the reaction?
Author Response
Reviewer 2
- Full names need to be given to abbrevations such as BDDT.
Brunauer-Deming-Deming-Teller (BDDT) (line 240). This was corrected in the Manuscript.
- What is the reason for Ba in BaCO3 but K in K2O from Fig1, when K(I) is more basic than Ba(II) and easier to form carbonate with CO2.
Recorded XRD spectra, in Figure 1, shows the 0.4Pt-3.5Ba-1.5K/Al2O3 catalyst pattern after the calcination process at 500 ºC in air. Residual carbonates are visible after the calcination and before the reaction-pretreatment phase. The signals associated with K4H2(CO3)3·1.5H2O phase (JCPDS 20-0886) would be overlapped with those corresponding to the more intense witherite phase of BaCO3.
- No XPS results was shown as claimed. It would be good if there was Cu XPS information.
XPS analysis results for the two Cu-zeolites are mentioned on line 293.
- The legend in Fig 6 is not clear.
This was corrected in the Manuscript (line 357).
- What is the main reason for the peak at 488 degC for Cu-SSZ-13 and at 420 for Cu-SAPO-34? What is the effect of stronger acid for Cu-SSZ-13 in the reaction?
High temperature acid sites refer to the Brønsted sites belonging to the zeolite framework structure without Cu. The incorporation of copper and the presence of different Cu-species produce a shift of the maximum NH3 desorption temperature at high T-range. The difference between the two isostructural zeolitic materials may be due to the zeolite framework composition. SAPO-34 zeolite contains Al, Si and phosphorus which increase the global acidity, while SSZ-13 is considered to be approximately a total-silica zeolite.
Strong acid sites for Cu-SSZ-13 would participate in the overall acidity of the zeolitic material, varying its ammonia retention capacity. The main active phase in the NH3-SCR reactions for the reduction of NOx to N2 in the NSR-SCR configuration employing alternating lean and rich cycles is related to the presence of the different Cu-species well-distributed and incorporated in the CHA-zeolite framework.

Round 2
Reviewer 2 Report
The authors just descried the XPS results without showing any figures as evidence. But where is the XPS? Without seeing the XPS result, such cliam should be considered not real. This is not acceptable.Either delete the claims about XPS or add XPS results figure is good.
Author Response
I am attaching the manuscript with the latest version, including a Table (Table 3) with XPS data that the reviewers had suggested.
Sincerely,